# Sentiment Analysis of Emirati Dialect

**Arwa A. Al Shamsi \*** and **Sherief Abdallah**

Faculty of Engineering and IT, The British University in Dubai, Dubai P.O. Box 345015, United Arab Emirates; sherief.abdallah@buid.ac.ae
\* Correspondence: 20180935@student.buid.ac.ae

**Abstract:** Recently, extensive studies and research in the Arabic Natural Language Processing (ANLP) field have been conducted for text classification and sentiment analysis. Moreover, the number of studies that target Arabic dialects has also increased. In this research paper, we constructed the first manually annotated dataset of the Emirati dialect for the Instagram platform. The constructed dataset consisted of more than 70,000 comments, mostly written in the Emirati dialect. We annotated the comments in the dataset based on text polarity, dividing them into positive, negative, and neutral categories, and the number of annotated comments was 70,000. Moreover, the dataset was also annotated for the dialect type, categorized into the Emirati dialect, Arabic dialects, and MSA. Preprocessing and TF-IDF features extraction approaches were applied to the constructed Emirati dataset to prepare the dataset for the sentiment analysis experiment and improve its classification performance. The sentiment analysis experiment was carried out on both balanced and unbalanced datasets using several machine learning classifiers. The evaluation metrics of the sentiment analysis experiments were accuracy, recall, precision, and f-measure. The results reported that the best accuracy result was 80.80%, and it was achieved when the ensemble model was applied for the sentiment classification of the unbalanced dataset.

**Keywords:** corpus; Emirati dataset; Arabic dialects; sentiment analysis; classification; classifiers

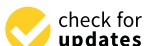



## 1. Introduction

Social networking has become one of the most popular communication platforms nowadays. People of different ages, cultures, and socioeconomic classes utilize social network environments to communicate a variety of messages to a worldwide audience [1,2]. Twitter, Instagram, and Facebook are examples of social networking platforms that allow users to communicate and freely discuss their thoughts and opinions in a non-constrictive atmosphere. This shared content can provide valuable data to companies [1], business owners, government authorities, services providers, and many institutes and organizations that may extract helpful ideas for their decision making and for the sake of improvements [3,4]. The study of thoughts, feelings, judgments, values, attitudes, and emotions regarding goods, services, organizations, persons, tasks, occasions, titles, and their attributes is known as a sentiment analysis [5]. A sentiment analysis involves a polarity classification task for recognizing positive, negative, or neutral texts to quantify what individuals believe using textual qualitative data [5].

Although there are a lot of studies on the analysis and classification of texts from social media sites, such as Twitter and Facebook, there are relatively few studies that use the Instagram platform to construct datasets and build resources for Arabic dialects [6].

Most previous research that used a sentiment analysis for the Arabic language focused on Modern Standard Arabic [7]. The most common dialects explored in sentiment analyses are Egyptian, Saudi, Algerian, Jordanian, Tunisian, and Levantine dialects [6]. However, to our knowledge, no researchers have used a sentiment categorization for the Emirati dialect.

In our most recent work, in [6,8], we reported that the majority of the constructed corpus in previous works that focused on the Arabic language either focused on the MSA

form or Arabic dialects but did not specify a specific dialect [6], or these studies focused on Twitter and Facebook [6]. Our study addresses this gap in the research by focusing on studying a specific Arabic dialect (of the United Arab Emirates), using text extracted from Instagram (which was shown to be the dominant social media website in the UAE [9]).

In this research paper, our main aim is to build an annotated corpus for Emirati dialect texts and evaluate the quality of the corpus using machine learning algorithms. Toward this aim, we followed the following steps.

First, we constructed a dataset from the Instagram platform. The constructed dataset originally consisted of more than 216,000 comments, mostly written in the Emirati dialect.

Second, we annotated the dataset based on the text polarity into positive, negative, and neutral categories. The number of annotated comments was 70,000.

Third, we applied preprocessing and TF-IDF features extraction to the constructed Emirati dataset.

Finally, we utilized different machine learning algorithms to conduct experiments on the constructed dataset using a sentiment analysis.

This research paper is organized as follows: Section 2 presents and surveys different research papers and articles that involve constructing a dataset for Arabic dialects and sentiment analysis experiments using machine learning algorithms. Section 3 describes the methodology and experimental setting. Section 4 presents the results and a discussion. Section 5 includes a conclusion and references.

## 2. Related Works

Research in the field of building resources and conducting sentiment analyses for Arabic dialects is increasing. Twitter and Facebook are the most commonly used platforms for building resources and constructing datasets of Arabic dialects. Moreover, most of the constructed datasets are manually annotated.

Table 1 below summarizes information about the constructed datasets for Arabic dialects in the most recent research papers. It is worth noting that the most used platform for constructing a dataset of the Arabic Dialects is Twitter followed by Facebook as illustrated in Figure 1 below.

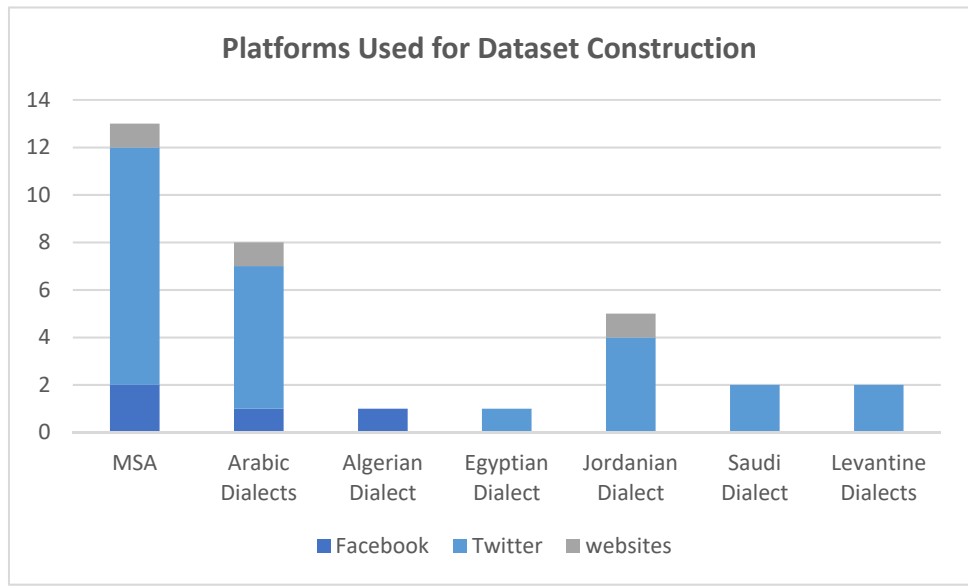

**Figure 1.** Platforms used for dataset construction.

**Table 1.** Datasets constructed for Arabic dialects.

| Ref. | Corpus Size | Platform | Dialect | Annotated (Yes/No) | Annotation Method |
|------|-------------|----------|---------|---------------------|-------------------|
| [3] | 7698 comments | Facebook | MSA and Algerian dialect | Yes | Manually |
| [10] | 8000 tweets | Twitter | Egyptian dialects and some other dialects | Yes | Lexicon-based annotation |
| [11] | 1103 tweets | Twitter | Arabic dialects | Yes | Manually |
| [12] | 1800 tweets | Twitter | MSA and Jordanian dialect | Yes | Manually |
| [13] | 2000 tweets | Twitter | MSA and Arabic dialects | Yes | Manually |
| [14] | 2071 Reviews | Apple Store and Google Play | MSA and Arabic dialects | Yes | Manually |
| [15] | 2242 tweets | Twitter | MSA and Levantine dialects | Yes | Lexicon-based annotation and manually |
| [16] | 15,274 reviews | Twitter, Facebook, Public Survey, and Mystery Shopper | Classical Arabic, MSA, and Arabic dialects | Yes | Manually |
| [12] | 151548 tweets | Twitter | MSA and Arabic dialects | Yes | Automatically |
| [17] | ARMD: 1000 reviews | elcinema.com cairo360.com | MSA and Arabic dialects | Yes | Lexicon-Based annotation |
| [18] | 8202 tweets | Twitter | MSA and Saudi dialect | Yes | Manually |
| [19] | 151,500 tweets | Twitter | MSA and Arabic dialects | Yes | Automatically |
| [20] | 22,550 tweets | Twitter | MSA and Jordanian dialect | Yes | Crowdsourcing Tool and Manually |
| [21] | 1000 tweets | Twitter | Jordanian dialect | Yes | Manually |
| [22] | 17,573 tweets | Twitter | MSA and Saudi dialect | Yes | Manually |
| [23] | 959 tweets | Twitter | MSA and Arabic dialects | Yes | Manually |
| [24] | 2730 reviews | JEERAN website | MSA and Jordanian dialect | Yes | Manually |
| [25] | 1798 tweets | Twitter | Levantine dialects | Yes | Manually |

It takes a long time to manually collect information about consumers' thoughts and sentiment data [15]. As a result, an increasing number of businesses and organizations are looking for automated sentiment analysis approaches to aid their understanding [15]. A sentiment analysis of some Arabic dialects may be challenging compared to MSA, which has computational tools and corpora and performs well in a sentiment analysis [15]. However, increased tools and classification models have recently been developed for Arabic dialects [15].

The approaches used for a sentiment analysis are machine-learning-based, knowledge-based, and hybrid approaches [24]. Machine-learning-based approaches involve building a classification model that learns from a labeled dataset. [26]. Knowledge-based approaches involve constructing and using lexicons of classified words [26]. Hybrid approaches are a combination of machine learning approaches and knowledge-based approaches. Table 2 includes all the abbreviations used in Table 3 which illustrates several studies and research that involved the utilization of machine learning algorithms for sentiment analysis of Arabic dialects.

**Table 2.** Abbreviations.

| | | |
|---|---|---|
| **Pos = Positive** | **DL = Deep Learning** | **BOW = Bag-of-Words** |
| **Neg = Negative** | NB = Naïve Bayes | ME = Maximum Entropy |
| **Neu = Neutral** | k-NN = K-nearest Neighbors | MNB = Multinomial NB |
| **Acc = Accuracy** | DT = Decision Tree | SGD -= SGD Classifier |
| **Pre = Precision** | RF = Random Forest | CNB = Complement Naïve Bayes |
| **Rec = Recall** | LR = Logistic Regression | PA = Passive Aggressive |
| **F = F-Measure** | CNN = Convolutional Neural Network | BNB = Bernoulli Naïve Bayes |
| **SVM = Support Vector Machine** | LSTM = Long Short-Term Memory | NSVC = Nu-Support Vector Classification |
| | RDG = Ridge Classifier | LSVC = Linear Support Vector Classification |

**Table 3.** Studies and research that involve sentiment analysis for Arabic dialects using machine learning algorithms.

| Ref. | Corpus Size | Platform | Dialect | Features | Classification Algorithms | Results | |
|---|---|---|---|---|---|---|---|
| [27] | 24,028 reviews | Arabic hotels' reviews (SemEval-ABSA16) | MSA and Arabic dialects | Morphological features/N -grams/ syntactic/semantic/ word embeddings | SVM—RNN | Acc = 95.4 | Prec = 94.48 |
| | | | | | | Rec = N/A | F = 93.4 |
| [28] | 2400 reviews | JEERAN website | MSA and Jordanian dialect | TF-IDF and n-grams | SVM | Acc = 93.25 | Pre = N/A |
| | | | | | | Rec = 91.92 | F = N/A |
| [29] | 11,647 tweets | Twitter | Saudi dialect | A cascade of multilayers | SVM/DT/K-NN/NB/DL | Acc = 85.25 | Pre = 85.30 |
| | | | | | | Rec = 88.41 | F = 86.81 |
| [30] | 2067 tweets | Twitter | MSA and Egyptian dialects | N-grams and POS (part of speech) | Lexicon-based/ SVM/NB | Acc = 96 | Pre = 92.8 |
| | | | | | | Rec = 90.2 | F = 91.5 |
| [31] | 1050 comments | Facebook | Sudanese dialect | Features from lexicon polarity | SVM/NB | Acc = 68.6 | Pre = 85.3 |
| | | | | | | Rec = 65.6 | F = 66.3 |
| [13] | 2000 tweets | Twitter | MSA and Arabic dialects | Feature vectors | SVM/NB/ME/ CNN/LSTM | Acc = 96 | Pre = 95 |
| | | | | | | Rec = 94 | F = 95 |
| [16] | 15,274 Reviews | Twitter, Facebook, Public Survey and Mystery Shopper | Classical Arabic, MSA and Arabic dialects | TF-IDF | Lexicon-based/ SVM/MNB/ SGD/LR | Acc = 95% | Pre = N/A |
| | | | | | | Rec = N/A | F = 93% |
| [32] | 8 datasets: 31,414 tweets | Twitter | MSA and Arabic dialects | Set of features extractedfrom text | CNB/SVM | Acc = 85.03% | Pre = N/A |
| | | | | | | Rec = N/A | F = N/A |
| [27] | 24,028 Reviews | Hotel's Reviews | MSA and Arabic dialects | Morphological features /n-grams/syntactic features/semantic features/word embedding | SVM/RNN | Acc = 95.4% | Pre = N/A |
| | | | | | | Rec = N/A | F = 93.4% |
| [33] | 2010 tweets | Twitter | Saudi dialect | TF-IDF | SVM, MNB, SGD, KNN, LR, PA | Acc = 75.9% | Pre = N/A |
| | | | | | | Rec = N/A | F = N/A |
| [34] | 63,257, reviews | goodreads.com | MSA and Arabic dialects | Bag-of-words | SVM, NB, DT, KNN | Acc = 57.8% | Pre = 70% |
| | | | | | | Rec = 57% | F = 63% |
| [35] | 151,548 tweets | Twitter | MSA and Arabic dialects | N-gram | RR/PA/NB/ SVM/BNB/ MNB/Stochastic Gradient De-cent/LR/ME/ Adaptive Boosting | Acc = 99.96% | Pre = 99.96% |
| | | | | | | Rec = 99.96% | F = 99.96% |

**Table 3.** *Cont.*

| Ref. | Corpus Size | Platform | Dialect | Features | Classification Algorithms | Results | |
|------|-------------|----------|---------|----------|--------------------------|---------|---|
| [36] | 1732 tweets | Twitter | MSA and Arabic dialects | TF/Tf-IDF/part of speech tagging/lexicon/Word2 vec | BNB/MNB/NSVC/LSVC/SGD/RGD/LR | Acc = 95% | Pre = N/A |
| | | | | | | Rec = N/A | F = N/A |
| [18] | 8202 tweets | Twitter | MSA and Saudi dialect | One-way ANOVA | SVM/NB/KNN/LR/MLP | Acc = N/A | Pre = N/A |
| | | | | | | Rec = N/A | F = 88% |
| [19] | 151,500, tweets | Twitter | MSA and Arabic dialects | TF | SVM/NB/BNB/MNB/SGD/LR | Acc = 93.5% | Pre = N/A |
| | | | | | | Rec = N/A | F = N/A |
| [21] | 1000 tweets | Twitter | Jordanian dialect | Feature vector uni-grams/bigrams/trigrams | SVM and NB | Acc = 82.1% | Pre = 85% |
| | | | | | | Rec = 84% | F = 84% |
| [23] | 959 tweets | Twitter | MSA and Arabic dialects | TF-IDF | SVM/NB/KNN | Acc = 88% | Pre = 100% |
| | | | | | | Rec = 98.34% | F = 88.08% |
| [37] | 17,000 comments | Facebook | MSA and Tunisian dialect | Doc2Vec | SVM/BNB/MLP | Acc = N/A | Pre = 80% |
| | | | | | | Rec = 98% | F = N/A |
| [38] | 3 datasets: 16,297 | Facebook and Twitter | MSA and Tunisian dialect | N-grams features | SVM and NB and lexicon-based approach | Acc = 94% | Pre = 93.9% |
| | | | | | | Rec = 93.8% | F = 93.9% |
| [24] | 2730 reviews | JEERAN website | MSA and Jordanian dialect | Features from lexicon | SVM | Acc = 95.6% | Pre = 96.02% |
| | | | | | | Rec = 95.07% | F = N/A |

## 3. Methodology and Experiment Setting

This section includes a detailed description of the methodology and steps of the experiment.

### 3.1. Dataset

Twitter, Facebook, Instagram, and YouTube are considered popular social networking platforms in the Arab world. Researchers stated that Twitter is the most popular social media platform in Gulf countries compared to the rest of the Arab world [10]. The researchers also found that, when compared to other Arab world nations, the Gulf countries have the lowest Facebook usage [10]. It was also found that most researchers used the Twitter and Facebook platforms to create Arabic text datasets [6].

The number of active social media accounts in the UAE amounted to about 20.85 million accounts, according to Alittihad newspaper, which reported that the number of active social media accounts in the UAE amounted to about 8.8 million Facebook accounts, 4 million LinkedIn accounts, 3.7 million Instagram accounts, 2.3 million Twitter accounts, and nearly 2 million active Snapchat accounts [9]. It is worth noting that Instagram witnessed an increase in active accounts in recent years [39]. Researchers stated that, in terms of active users, Instagram is the third most popular social networking platform [39]. It is also the most popular social media platform among teenagers, as well as the most popular platform for influencer marketing.

Instagram is a well-known photo- and video-sharing social media platform. It is one of the most widely used social media platforms in the UAE. People can leave comments on Instagram posts, as well as like or dislike the photographs and videos that have been posted [8]. We focus here on gathering comments on Instagram posts written in Arabic dialects, with a particular concentration on the Emirati dialect due to many reasons: (a) the authors' goal is to collect texts written in the Emirati dialect from popular Emirati Instagram accounts. Based on our findings while investigating Twitter, Facebook, and Instagram, we discovered that the number of comments left on the same posts by the same account owners on Instagram was much higher than the number of comments left on the same posts by the same account owners on Twitter and Facebook. (b) According to [40], Instagram is one of the top three social media platforms in the United Arab Emirates; Instagram is popular social media site in UAE and its popularity is increasing day after day. (c) Finally, while

reviewing research papers, most of the constructed datasets used Twitter and Facebook platforms to construct datasets of Arabic texts; however, a limited number of studies targeted the Instagram platform for collecting comments and constructing datasets from the collected comments [6].

Below is our methodology, which was followed for collecting the dataset:

1.  We were granted a "Facebook for Developers" account, through which we were able to collect comments from Instagram posts written in Arabic.
2.  We identified public Instagram accounts of Emirati government authorities, Emirati news accounts, and Emirati bloggers to collect comments written in the Emirati dialect.
3.  Comments in the form of Arabic texts were collected from different kinds of posts whether they were pictures or videos. The collected comments were stored in an Excel datasheet.

We were able to initially collect around 216,000 comments; after that, around 70,000 comments were annotated and categorized as positive, negative, or neutral. Moreover, the comments were annotated based on dialect type: either the Emirati dialect or other Arabic dialects. The criterion for the annotation will be further mentioned below.

It was reported that most of the comments were written in the Emirati dialect; however, some comments were written in MSA and other Arabic dialects. In total, 459 comments were written in other Arabic dialects, 1 comment was written in MSA, and the remaining 69,540 comments were written in the Emirati dialect. No occurrences of Arabic writing with Roman characters were reported in the dataset. Table 4 illustrates a description of our collected corpus of the Emirati dialect. We named our corpus ESAAD, which stands for Emirati Sentiment Analysis Annotated Dataset.

**Table 4.** Description of Emirati Sentiment Analysis Dataset (ESAAD).

| Properties | Positive | Negative | Neutral |
|---|---|---|---|
| Number of Comments | 27,588 | 23,946 | 18,466 |
| Number of Words | 132,070 | 233,945 | 110,622 |

Figure 2 illustrates a comparison of the sentiment score values of Instagram comments in terms of the number of neutral, positive, and negative comments.

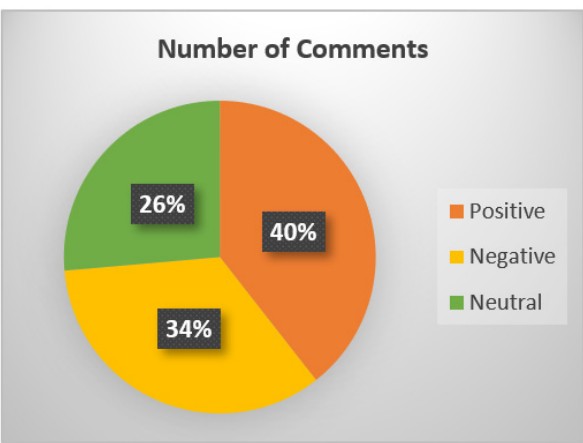 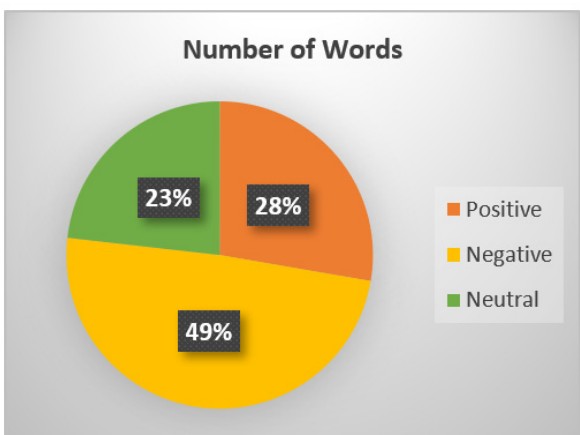

**Figure 2.** A comparison of the sentiment score values of Instagram comments and words in terms of the number of neutral, positive, and negative comments.

Ethics: The comments gathered are from public accounts, and this action is lawful and approved under the website's Terms of Use Policies. The collected comments are in the form of small sentences that are not protected by copyright requirements, and to ensure adherence to the privacy rules, the identities of account owners, from which the texts were obtained, are disguised.

*3.2. Annotating the Dataset*

A sentiment annotation of a text involves labeling the text based on the sentiment presented in that text. Three approaches can be utilized for sentiment annotation: automatic annotation, semi-automatic annotation, and manual annotation [41]. In this research paper, the constructed dataset was manually annotated by three annotators. Each of the annotators agreed on a set of constraints and signed a contract with the author. Each annotator was given an annotation guideline to follow, which was based upon annotation principles developed by V. Batanović, M. Cvetanović, and B. Nikolić [42]. The annotation guideline principles are as follows:

- The annotator must enter the term "positive" for the comment, which expresses a positive sentiment. Similarly, if the comment expresses a negative sentiment, the annotator must enter the term "negative", which signifies a negative sentiment. If the comment does not represent any sentiment, then the annotator must enter the word "neutral".
- Each comment is individually evaluated, with no recourse to the surrounding comments. The only difficulty that the annotators faced mostly revolved around the uncertainty about whether a given comment is sarcastic. Researchers dealt with this difficulty by approving the sentiments of certain comments that the majority of annotators agreed on.
- A predefined list with several positive, negative, and neutral words was identified and handed to annotators to help them in their annotations.
- The researchers used a composite scoring system when a comment had several different statements. This indicates that each sentence was independently assessed, with the final sentiment label derived by merging the partial labels. In this experiment, the strength of the sentiment is not considered, and texts of a certain polarity had the same sentiment label regardless of the strength of the polarity they represented.
- For sentiment labeling, only comments in which the author was the speaker were considered. Other people's points of view indicated in the comment were only taken into account if they indirectly exposed the speaker's position; otherwise, they were viewed as neutral. Comments that presented questions, requests, advertisements, follow requests, suggestions, pleas, intent, or ambiguous statements were labeled as neutral sentiments. Comments that represented news, factual statements, or any kind of information were labeled as neutral sentiments. Comments that represented regrets and sarcasm were labeled as negative sentiments. Humorous comments were labeled as positive sentiments, as humor indicates happiness, unless this humor was followed by any negative words, then the sentiment of the sentence was calculated based upon the composite scoring system mentioned earlier. Emojis were considered when annotating comments, and the weights given to the emoji's polarity were less than the weights given to the text's polarity as this method could deal with the weaknesses that were related to the misuse of emojis [1]. It was found that emoticons or emojis could lead to several errors while analyzing the sentiment of texts, as they were more likely to be used in social media texts that utilized emojis or emoticons that contradicted the text [43]. Comments that contained verses from the Holy Qur'an or hadiths from the Sunnah were labeled with neutral sentiments.

The annotation process:

1. Each comment was annotated by the three annotators, then the sentiment for the comment was approved by determining the sentiment that the majority of annotators agreed on.
2. If there was no consensus between the three annotators, another expert annotator was assigned.
3. Annotators successfully annotated 70,000 comments that were extracted from the Instagram platform.

4. To measure the quality of annotation, an inter-annotator agreement was calculated for 3000 selected comments. The authors utilized Cohen's kappa coefficient to measure the agreement between the annotators using SPSS software.

It is worth noting that SPSS software restricted the following rules for correctly implementing Cohen's kappa coefficient: (1) The raters' responses were graded on a nominal scale, and the categories must be mutually exclusive (in our experiment: positive, negative, or neutral). (2) The response data were made up of paired observations of the same phenomena, which meant that both raters evaluated the identical observations (in our experiment, annotators evaluated the same comments, i.e., text). (3) The two raters were independent, which meant that one rater's decision did not influence the decision of the other (in our experiment, the annotators were independent). (4) All observations were judged by the same raters, meaning that the raters were fixed or unique (in our experiment, the annotators were fixed, meaning that the same annotators evaluated the same text).

For example, both annotators evaluated the same 3000 comments, i.e., text. Additionally, the ratings of the annotators (i.e., either "Positive", "Negative" or "Neutral" sentiments) were compared for the same comments (i.e., text; the rating given by annotator 1 for comment 1 was compared to the rating given by annotator 2 for comment 1, and so forth).

Cohen's kappa coefficient was utilized to measure the inter-annotator agreement, and the result was 93%.

### 3.3. Preprocessing

The preprocessing stage is defined by [10] as the process of cleaning data to decrease errors and improve sentiment analysis performance. The preprocessing phase is critical for reducing noise and improving sentiment analysis results. Moreover, text preprocessing is an essential stage for developing any word embedding model since it may have a big impact on the final results [44]. Preprocessing involves several steps, such as tokenization, diacritics removal, non-Arabic words and letters removal, punctuation removal, normalization, stemming, and stop words removal [45]. A recent study [5] reviewed the different preprocessing steps applied for the sentiment mining of Arabic dialects and found the following: (a) In most studies, the data cleaning step was applied, which involved removing URLs, diacritics, hashtags, punctuation, and special characters. (b) Stop words removal was an important step in preprocessing stage. (c) After the data cleaning step and stop words removal, the normalization step took place, was implemented in most of the explored studies and involved replacing the Arabic letters (آ إ أ ٱ) with (ا), replacing (ة) with (ه), replacing (ئ ي) with (ي), and replacing (ؤ) with (و); moreover, in some studies, the emoticons were replaced in the normalization step. (d) Normalization was challenged by the various ways that a single word could be written in a dialect; as a result, most researchers used stemming for structured comments and discarded unstructured comments. (e) The biggest challenge of handling an Arabic dialect is the automatic generation of the stop word dictionary and the normalization of data by looking for the roots of words [5]. In this research paper, the authors implemented the following preprocessing steps using the NLTK (Natural Language Toolkit) library in Python:

- Tokenization: This is an initial step of preprocessing, and it involves breaking up the text into words (tokens) separated by white spaces or punctuation [13]. This operation yields a set of words as a result. In this experiment, tokenization was implemented using the NLTK library.
- Non-Arabic content filtering: Filtering non-Arabic content from the collected dataset was the initial stage in the preprocessing stage. This phase is critical, especially when dealing with data from the "web," such as the data from "Instagram", in our experiment. Although Arabic is easily recognizable by its alphabet, it is worth mentioning that Arabic letters are also used in other languages, such as Urdu and Persian [44]. To recognize the Arabic language, a Python library for language detection was utilized.

- Data cleaning: This step involves the elimination of Arabic diacritics, which seldom appear in Arabic text and are often regarded as noise; this process is referred to as diacritics removal or sometimes dediacritization. Short vowels, shadda (gemination marker), and dagger alif are among these diacritics. Arabic diacritics include ( اَ, and these symbols may appear above or under the Arabic letters. In this experiment, Arabic diacritics were removed (for example, "مُتَمَيِّزَ" was returned to its original form "متميز"). Moreover, data cleaning involves unnecessary character removal, i.e., characters that have no phonetic value are removed [46]. In this experiment, the authors removed the tatweel (kashida) character, and numbers were also removed. Moreover, all symbols and other unknown characters were removed, such as punctuation. Furthermore, URLs and hashtags were removed. It is worth noting that the text of the hashtag remained in the text and was not removed because, in most cases, hashtags represent a sentiment that can be taken into consideration.

- Stop words removal involved the elimination of words that are used to structure language but do not add to its content in any manner [5]. Some of these terms include (هذه ، هذا ، هؤلاء ، الذين). It is worth mentioning that this step is very important [47].

- Normalization: This involved replacing Arabic letters (أ إ آ) with (ا), replacing (ة) with (ه), replacing (ئ ي) with (ي), and replacing (ؤ) with (و). Elongated words were also returned to their original form (for example, "حلوووووو" was returned to its original form "حلو").

- Stemming: stemming methods are an important part of the preprocessing step. The different words that arose from the same word were mapped to their root or stem once the irrelevant information was removed. Stemming is a preprocessing technique that reduces the high dimensionality of vector space by reducing a word to its root or stem [48]. A stemmed term has a larger meaning than the original word, and it can help you save space. Stemming is a technique for improving a sentiment analysis by reducing derived or inflected words to their stems, bases, or root form, which aids in grouping all variations of a word into a single category, reducing entropy, and providing a better indication of data [21]. In [13], researchers investigated the impact of stemming on sentiment classification and reported that light stemming methods outperformed root extraction methods [13]. Light stemming maintains the meaning of information by deleting just the suffix and prefix terms. However, because this root-extraction method involves extracting the root or base of each word, it misses out on some important morphological information [13]. In this study, stemming was accomplished by eliminating any associated suffixes and prefixes from words in Instagram comments as the author utilized a light stemming approach to stemming.

### 3.4. Features Extraction

The feature extraction step is essential as it reduces dimensionality; therefore, the computational cost is also reduced. Furthermore, feature extraction benefits avoiding the learning model's overfitting to the training data [16].

Machine learning algorithms only deal with numeric data; hence, the input data must be transformed into numeric vector format for text categorization. In general, this operation may be accomplished in one of two ways: CountVectorizer or TfidfVectorizer [49]. The CountVectorizer involves a simple operation of word count. The TfidfVectorizer, on the other hand, uses the Tf-IDf score as numerical data for the vector model [49].

In our experiment, we utilized the term frequency-inverse document frequency (TF-IDF) feature matrix, which showed the significance of terms in a review of the corpus [16]. Term frequency represents how often a given word appears within a text, while Inverse Document Frequency adjusts words that appear so many times in texts [50]. We utilized SciKit-Tfidf Learn's vectorizer Python package for this, which also ignored phrases with a document frequency more than or less than a minimum and maximum threshold, respectively [16].

*3.5. Sentiment Analysis Experiment Setup*

We used the following classifiers after converting the dataset into TfidfVectorizer: logistic regression (LR), multinomial Naïve Bayes (MNB), support vector machine (SVM), decision tree (DT), random forest (RF), multilayer perceptron (MLP), AdaBoost, GBoost, and an ensemble model. SciKit-Learn (SKLearn) library and Natural Language Tool Kit (NLTK) library were utilized in the experiment. These libraries are open-source and free, allowing programmers to use a variety of machine learning techniques.

As mentioned earlier, the dataset consisted of 70,000 comments extracted from the Instagram platform, and most of the comments were written in the Emirati dialect. In the experiment, the dataset was divided into a train set, test set, and validation set.

We used the pandas API in Python to upload the dataset (.csv file), and then used the TfidfVectorizer to train the dataset. Various classification experiments were conducted on the dataset to compare the classification results obtained using different machine learning algorithms. We used 80% of the dataset as training data, 10% of the dataset as test data, and 10% of the dataset as a validation set. All of the experiments were conducted using the TF-IDF feature referred to in Section 3.4. Figure 3 illustrates the experiment model.

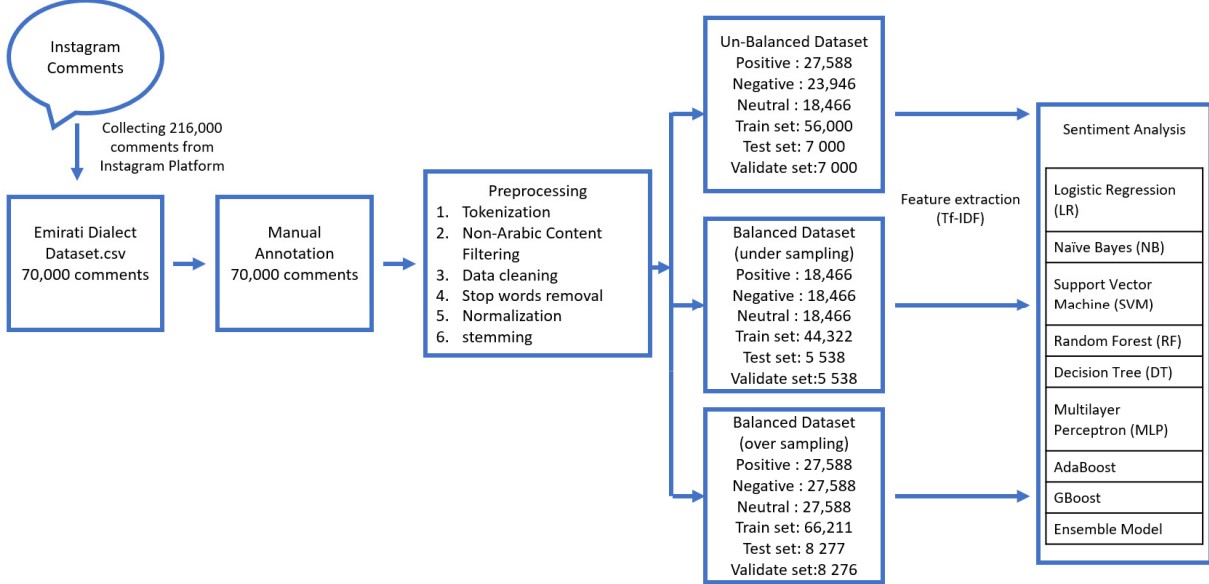

**Figure 3.** Sentiment analysis experiment model.

We utilized machine learning algorithms for both balanced and unbalanced datasets. In order to handle the unbalanced dataset and convert it into a balanced dataset, both undersampling and oversampling techniques were applied.

Undersampling involves reducing the number of majority target instances or samples., i.e., reducing the number of comments in each sentiment class to reach the minority class instances [51]. Oversampling, on the other hand, created new examples of existing comments to increase the number of minority class instances or samples [51].

Further details about the utilized classifiers and the dataset are described below.

Logistic regression (LR) is a linear classifier that uses a straight line to try to differentiate between positive and negative cases [52]. LR is one of the most elegant and widely used classification algorithms and it generally does not overfit data. LR is commonly trained using the gradient descent optimization approach to obtain the model parameters or coefficients [52].

Multinomial naïve Bayes (MNB): The naïve Bayes NB theorem is the foundation of the multinomial naïve Bayes classifier. Although NB is extremely efficient, its assumptions have an impact on the quality of the findings [50]. The multinominal naïve Bayes (MNB) approach was developed to overcome NB disadvantages. MNB uses a multinomial model

to represent the distribution of words in a corpus. MNB treats the text as if it were a series of words, assuming that the placement of words is independently determined of one another [50].

Support vector machine (SVM): The SVM algorithm is a supervised technique that may be used for both classification and regression [49]. By identifying the nearest points, it calculates the distance between the differences between the two classes [49].

Random forest (RF): The random forest classifier is a supervised technique that may be used for classification and regression [49]. It is a collection of several independent decision trees put together as a whole [53]. It makes a decision based on the results of numerous decision trees with the maximum scores [49].

Multilayer perceptron (MLP): Multi-layer perceptron is a popular artificial neural network (ANN). It is a strong modeling tool that uses examples of data with known outcomes to apply a supervised training approach. This approach creates a nonlinear functional model that allows output data to be predicted from input data [54].

Decision tree (DT): One of the most often utilized approaches in classification is the decision tree algorithm. A DT's goal is to create a model that uses input data to predict the correct label for target variables [50]. It is a supervised method that builds a tree to hierarchically build a model. Each decision node in the tree is connected with an attribute [53]. A decision node has two or more branches, each with an associated attribute value or a range of values. The target value is contained in a leaf node, which has no branches [53]. To achieve maximal homogeneity at each node, the algorithm divides the training data into smaller sections. The method starts at the root node and works its way down the tree using the decision rules specified for each decision node to obtain the outcome for a sample [53].

AdaBoost: AdaBoost is a boosting algorithm that creates a resilient model that is less biased than its constituents by employing a large number of weak learners (base estimators) [53]. The base estimators are trained in a way that each base model is dependent on the prior one, and the predictions are merged using a deterministic approach [53]. When fitting each base model, additional weight was assigned to samples that were handled inaccurately by previous models in the sequence. A strong learner with little bias was obtained at the end of the procedure [53].

GBoost: GBoost is a boosting algorithm, as previously discussed. The gradient boosting (GBoost) approach employs an iterative optimization procedure to create the ensemble model as a weighted sum of several weak learners, similar to AdaBoost [53]. The distinction derives from the sequential optimization process' definition [53]. The GBoost technique uses gradient descent to describe the issue. At each step, a base estimator is fitted opposite to the gradient of the current ensemble's error curve [53]. GBoost utilizes gradient descent, whereas AdaBoost tries to solve the optimization issue locally at each step [53].

Ensemble Model: Ensemble learning is a type of machine learning that involves training several different classifiers, and then selecting a few to use in an ensemble [55]. It has been proven that combining classifiers is more successful than using each one separately [55]. In our experiment, we utilized an ensemble model of RF, MNB, LR, SMV, and MLP, and it presented the best result in terms of accuracy.

Dataset description: Table 5 summarizes the balanced and unbalanced datasets.

**Table 5.** Summary of the balanced and unbalanced datasets.

| | Total Size | Positive | Negative | Neutral | Train Set | Test Set | Validation Set |
|---|---|---|---|---|---|---|---|
| **Unbalanced dataset** | 70,000 | 27,588 | 23,946 | 18,466 | 56,000 | 7000 | 7000 |
| **Balanced dataset (under sampling)** | 55,398 | 18,466 | 18,466 | 18,466 | 44,322 | 5538 | 5538 |
| **Balanced dataset (oversampling)** | 82,764 | 27,588 | 27,588 | 27,588 | 66,211 | 8277 | 8276 |

1.　　Description of the unbalanced dataset:

Total positive comments: 27,588; total negative comments: 23,946; total neutral comments: 18,466; train set: 56,000; test set: 7000; and validation set: 7000.

2.　　Description of the balanced dataset (undersampling):

Dataset size: 55,398 comments (18,466 positive comments; 18,466 negative comments; 18466 neutral comments);

Train set: 44,322 comments (14,774 positive comments; 14,774 negative comments; 14774 neutral comments);

Test set: 5538 comments (1846 positive comments; 1846 negative comments; 1846 neutral comments);

Validation set: 5538 comments (1846 positive comments; 1846 negative comments; 1846 neutral comments).

3.　　Description of the balanced dataset (oversampling):

Dataset size: 82,764 comments (total positive comments: 27,588; total negative comments: 27,588; total neutral comments: 27,588);

Train set: 66,211, Test set: 8277, and validation set: 8276.

## 4. Results and Discussion

We utilized Python and its text mining libraries to create and implement our framework. The following tables and figures present the results of the experiments conducted using various classifiers. The performance evaluation was carried out using four key metrics:

The accuracy represents the "number of correct predictions made divided by the total number of predictions" [33]. Accuracy shows the percentage of correctly categorized texts, regardless of class [34]:

- Recall: represents the "number of correct positive results divided by the total number of positive results that should have been returned" [33]. A great recall indicates that a large number of comments from the same class are correctly categorized [34].
- Precision: "it is the number of correct positive results divided by the total number of positive results" [33]. The greater the precision percentage, the more precise the positive class prediction [34].
- F1-score: F-score or F-measure are other terms for the F1-score. The F1-score is the average of the precision (the number of correct positive results divided by the total number of positive results) and recall (the number of correct positive results divided by the total number of positive results that should be returned), with 1 being the best and 0 being the worst [33].

The sentiment analysis experiments were conducted on both balanced and unbalanced datasets. Additionally, the experiments were conducted on the balanced dataset after undersampling and oversampling. Eight machine learning classifiers were utilized for the sentiment analysis experiments on balanced and unbalanced datasets. Moreover, an ensemble model, in which several classifiers were combined, was used. The results of the sentiment analysis experiments are presented in Tables 6–8. Additionally, the results are also illustrated in Figures 4–6.

**Table 6.** Classification results for sentiment analysis of unbalanced dataset.

| | Unbalanced Dataset (Features Extraction: TF-IDF) | | | |
|---|---|---|---|---|
| | Accuracy | Recall | Precision | F-Measure |
| **Logistic Regression (LR)** | 79.87% | 65.15% | 78.48% | 67.34% |
| **Decision Tree (DT)** | 73.85% | 67.34% | 66.69% | 66.99% |
| **Multinomial Naïve Bayes (MNB)** | 77.21% | 56.93% | **86.71%** | 55.54% |
| **Support Vector Machine (SVM)** | 79.74% | 68.54% | 74.89% | 70.44% |
| **Random Forest (RF)** | 79.17% | **70.74%** | 73.22% | **71.56%** |
| **Multilayer Perceptron (MLP)** | 79.74% | 69.64% | 74.44% | 71.27% |
| **AdaBoost** | 76.91% | 60.95% | 71.63% | 62.06% |
| **GBoost** | 76.34% | 60.53% | 69.99% | 61.52% |
| **Ensemble Model** | **80.80%** | 68.27% | 78.04% | 70.66% |

Note: the best performance results are bolded.

**Table 7.** Classification results for sentiment analysis of balanced dataset (undersampling).

| | Balanced Dataset (Undersampling) (Features Extraction: TF-IDF) | | | |
|---|---|---|---|---|
| | Accuracy | Recall | Precision | F-Measure |
| **Logistic Regression (LR)** | 70.41% | 70.41% | 70.39% | 70.03% |
| **Decision Tree (DT)** | 59.80% | 59.80% | 60.21% | 59.03% |
| **Multinomial Naïve Bayes (MNB)** | 65.59% | 65.59% | 67.98% | 65.79% |
| **Support Vector Machine (SVM)** | 70.66% | 70.66% | 70.61% | 70.37% |
| **Random Forest (RF)** | 66.44% | 66.44% | 67.59% | 66.01% |
| **Multilayer Perceptron (MLP)** | 69.36% | 69.36% | 69.38% | 69.22% |
| **AdaBoost** | 53.62% | 53.62% | 59.91% | 53.17% |
| **GBoost** | 57.94% | 57.94% | 65.31% | 57.22% |
| **Ensemble Model** | **70.97%** | **70.97%** | **70.80%** | **70.72%** |

Note: the best performance results are bolded.

**Table 8.** Classification results for sentiment analysis of balanced dataset (oversampling).

| | Balanced Dataset (Oversampling) (Features Extraction: TF-IDF) | | | |
|---|---|---|---|---|
| | Accuracy | Recall | Precision | F-Measure |
| **Logistic Regression (LR)** | 75.48% | 75.48% | 75.54% | 75.42% |
| **Decision Tree (DT)** | 70.14% | 70.14% | 71.17% | 69.87% |
| **Multinomial Naïve Bayes (MNB)** | 76.25% | 76.25% | 76.28% | 76.19% |
| **Support Vector Machine (SVM)** | 76.11% | 76.11% | 76.18% | 76.07% |
| **Random Forest (RF)** | 73.84% | 73.84% | 74.77% | 73.71% |
| **Multilayer Perceptron (MLP)** | 76.51% | 76.51% | 76.95% | 76.44% |
| **AdaBoost** | 53.54% | 53.53% | 60.64% | 53.08% |
| **GBoost** | 57.69% | 57.69% | 64.89% | 57.27% |
| **Ensemble Model** | **77.51%** | **77.51%** | **77.66%** | **77.48%** |

Note: the best performance results are bolded.

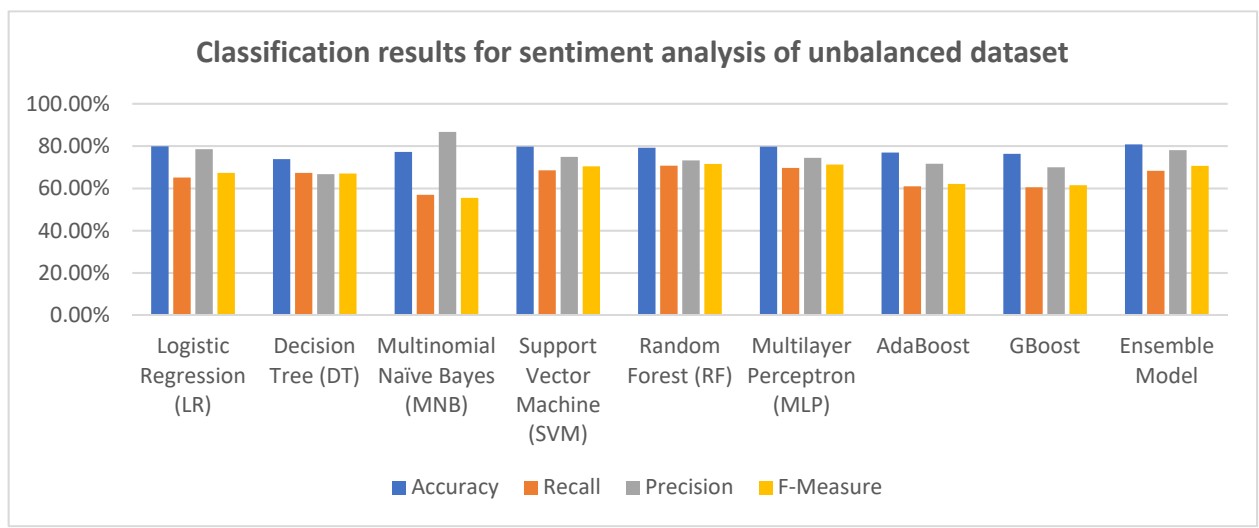

**Figure 4.** Classification results for sentiment analysis of unbalanced dataset.

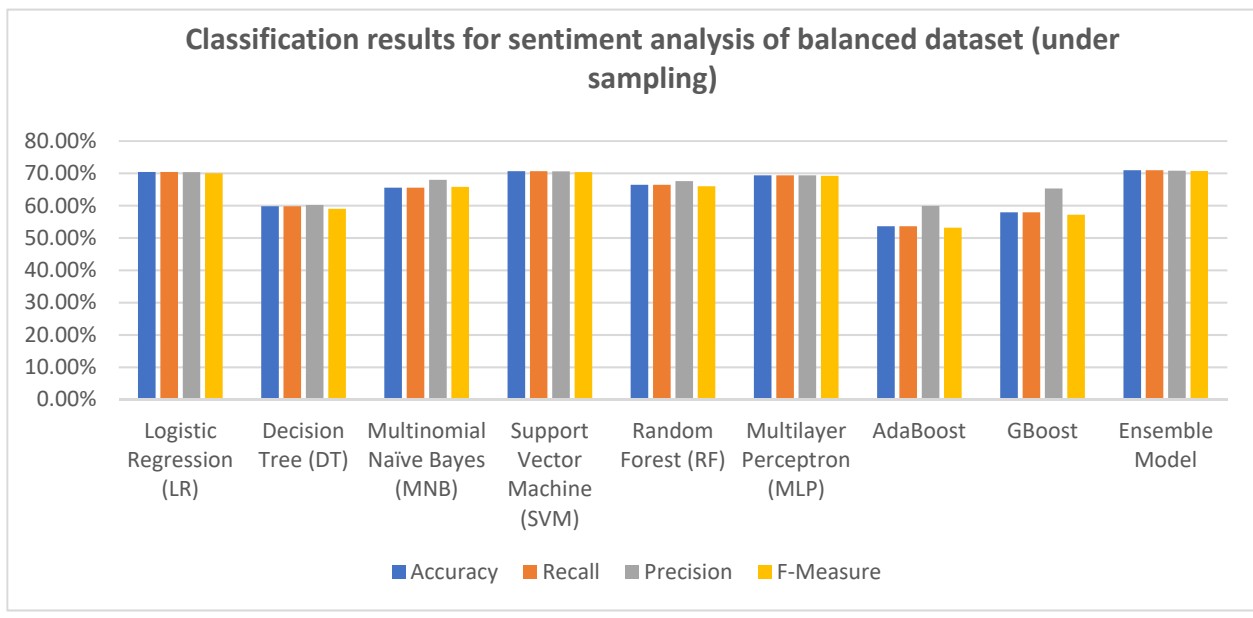

**Figure 5.** Classification results for sentiment analysis of balanced dataset (undersampling).

From the experimental results shown in Table 4 and Figure 3, we compared the results achieved when using different machine learning algorithms for the unbalanced dataset. The results revealed that the ensemble model, which combines RF, MNB, LR, SMV, and MLP algorithms, outperformed all other classifiers in terms of accuracy 80.80%. Moreover, the random forest classifier presented the best result in terms of recall and F-measure, and the multinomial naïve Bayes presented the best result in terms of precision. On the other hand, the lowest performance in terms of accuracy and precision was achieved when using the decision tree classifier (accuracy = 73.85%) and (precision= 66.69%).

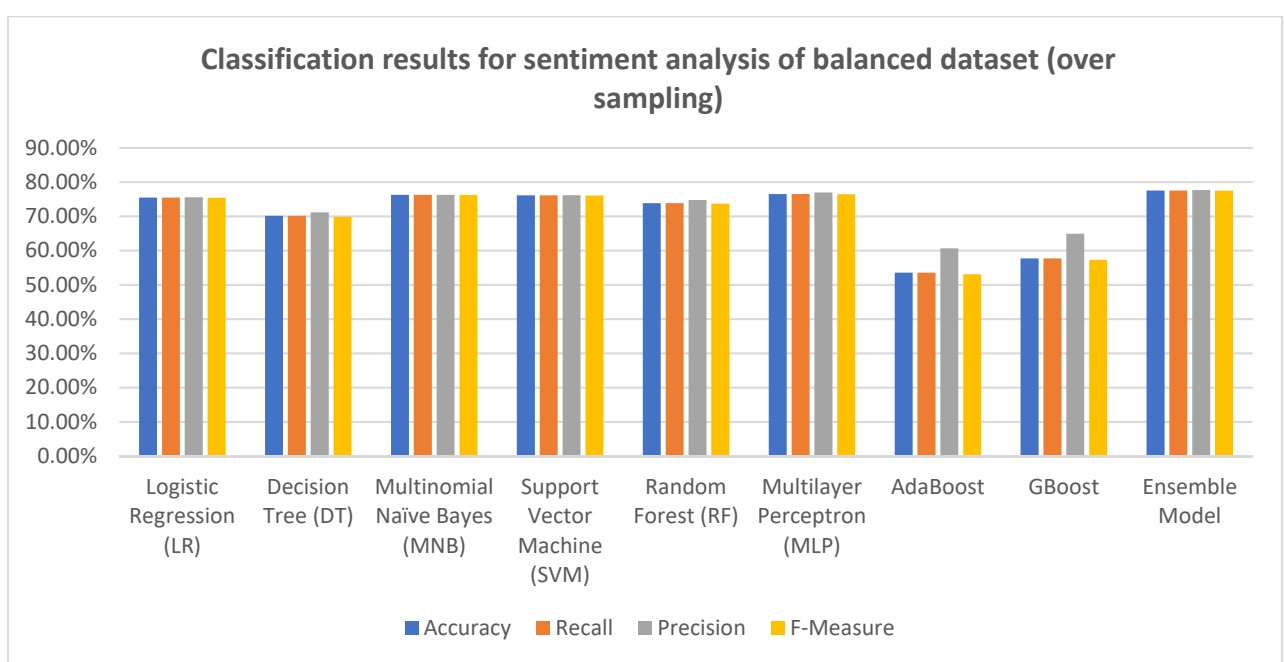

**Figure 6.** Classification results for sentiment analysis of balanced dataset (oversampling).

From the experimental results shown in Table 5 and Figure 4, we compared the results achieved when using different machine learning algorithms on the balanced dataset after undersampling. The results revealed that the ensemble model, which combines RF, MNB, LR, SMV, and MLP algorithms outperformed all other classifiers in terms of accuracy (70.97%), recall (70.97%), precision (70.80%), and F-measure (70.72%). On the other hand, the least performance in terms of accuracy, recall, precision, and F-measure was achieved when using the AdaBoost classifier. It is worth noting that SVM performance was the best among the other used machine learning classifiers in terms of accuracy, recall, precision, and F-measure; this result aligned with the result achieved by [21,23], where SVM performance was the best.

From the experimental results shown in Table 6 and Figure 5, we compared the results achieved using different machine learning algorithms on the balanced dataset after oversampling. The results revealed that the ensemble model, which combines RF, MNB, LR, SMV, and MLP algorithms, outperformed all other classifiers in terms of accuracy (77.51%), recall (77.51%), precision (77.66%), and F-measure (77.48%). On the other hand, the lowest performance in terms of accuracy, recall, precision, and F-measure was achieved when using the AdaBoost classifier.

When comparing the results from different experiments, it was reported that the best results in terms of accuracy, recall, precision, and F-measure were achieved when using different machine learning algorithms on the unbalanced dataset, and the best performance in terms of accuracy was achieved by utilizing an Ensemble model on the unbalanced dataset (accuracy = 80.80%). This result aligned with the findings reported in [16], that the machine learning classifier performance on the unbalanced dataset was better than the performance on the balanced dataset.

It was also reported that the performance of different classifiers in the oversampling dataset outperformed the performance in the under sampling dataset.

It is worth noting that, despite the high performance of the machine learning classifiers in the dataset text classification, the misclassification of text polarity is a challenge. In this research paper, we reported the following reasons for the misclassification of the text in the dataset of the Emirati dialect: (a) the text may include multiple polarities; (b) negation is challenging; and (c) the dialects do not have standard orthographic written form, meaning that the same word may be written in many different ways and may include spelling

errors as well. (d) It was reported by annotators that, in many comments, the emojis used represent sentiments different from the text sentiments, and (f) polysemy is a challenge as the same word may have different meanings.

## 5. Conclusions

We created the first manually annotated Instagram dataset for a sentiment analysis of the Emirati dialect in this research paper. The constructed dataset consisted of 70,000 comments, the majority of which were written in the Emirati dialect. We assessed the quality of our corpus using Cohen's kappa coefficient, which revealed it to be of high quality. Furthermore, we evaluated the quality of the collected corpus by applying eight different machine learning techniques and measuring their performance. For text vectorization, we used TF-IDF. The results show the corpus has a high quality with many techniques, achieving more than a 70% accuracy (the highest accuracy achieved was 80%).

**Author Contributions:** Conceptualization, methodology, validation, formal analysis, investigation, resources, data curation, writing—original draft preparation, visualization, project administration, and funding acquisition, A.A.A.S.; writing—review and editing, supervision, S.A. All authors have read and agreed to the published version of the manuscript.

**Funding:** This research received no external funding.

**Data Availability Statement:** The dataset used in this research paper is constructed by the authors and it will be available for the research community, for dataset inquiry please contact the author via email: 20180935@student.buid.ac.ae.

**Acknowledgments:** This work is a part of a project submitted to The British University in Dubai.

**Conflicts of Interest:** The authors declare no conflict of interest.

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
