# Peer review of "Sentiment Analysis of Emirati Dialect"

_2504-2289, doi:10.3390/bdcc6020057_

Round 1
Reviewer 1 Report
p. 1, row 8: *,moreover, > . Moreover,
pp. 2-3, passim: the formula and the constructed dataset was annotated is repeated 12 times! I think, you should try to avoid such repetitions.
p. 11: the description of the dediacritization process is interesting. However, you should indulge more time and space to the following points:
a) how the specific color of the Emirati dialect appears in the corpus?
b) are there occurrences of Arabic writing with Roman characters?
c) is all your corpus of a dialectal nature or is there an oscillation bewteen Standard Arabic and Emirati dialect (perhaps through the intermediate register of wusta?)
d) I think, you should bring some more examples. The examplification through the issue of dediacritization is interesting but insufficient.
Author Response
p. 1, row 8: *,moreover, > . Moreover, (Done)
pp. 2-3, passim: the formula and the constructed dataset was annotated is repeated 12 times! I think, you should try to avoid such repetitions. ( Done)
p. 11: the description of the dediacritization process is interesting. However, you should indulge more time and space to the following points:
a) how the specific color of the Emirati dialect appears in the corpus? ( the corpus is annotated for sentiment and dialect type, the Dialects are Emirati, Arabic, and MSA) it is well described on page 9
b) are there occurrences of Arabic writing with Roman characters? (no and it is mentioned in page 9)
c) is all your corpus of a dialectal nature or is there an oscillation bewteen Standard Arabic and Emirati dialect (perhaps through the intermediate register of wusta?) (mentioned in page 9)
Reviewer 2 Report
This is an interesting paper and the methodology as well as the explanation for choosing this methodology is presented in a clear way and it follows standards in the field, as expected. The authors go into a lot of detail to explain the reasons behind their thinking and to provide accurate definitions of all different components they are using. However, the paper suffers from poor English syntax, and I cannot recommend it for publication at its current form. The authors should have the text proofread and then resubmit.

Author Response
Major Comments
I don’t see the purpose of the first paragraph in Section 2. The information is summarized in the table, so I recommend reducing this paragraph, which reads like a long list of notes rather than a literature review. (Done) page 2 and 3
While I find the use of bullet points helpful to the reader, there are too many of them in section 3.2. After line 210, I don’t see the need for bullet points as the text is explanatory to the process and could be in a paragraph. Lines 218-237 could be shown with a table instead. (Done) page 11
The author(s) do not report misclassification and if there are any, what type of
misclassifications are they? How do they interpret them? (Done) page 20
Minor Comments
Please look for these kind of typos throughout the text since I did not mark all.
all the typos comments fixed (Done)